# Synthesis, Molecular Docking Study, and Cytotoxicity Evaluation of Some Novel 1,3,4-Thiadiazole as Well as 1,3-Thiazole Derivatives Bearing a Pyridine Moiety

**DOI:** 10.3390/molecules27196368

**Published:** 2022-09-27

**Authors:** Amr S. Abouzied, Jehan Y. Al-Humaidi, Abdulrahman S Bazaid, Husam Qanash, Naif K. Binsaleh, Abdulwahab Alamri, Sheikh Muhammad Ibrahim, Sobhi M. Gomha

**Affiliations:** 1Department of Pharmaceutical Chemistry, College of Pharmacy, University of Hail, Hail 81442, Saudi Arabia; 2Department of Pharmaceutical Chemistry, National Organization for Drug Control and Research (NODCAR), Giza 12311, Egypt; 3Department of Chemistry, College of Science, Princess Nourah Bint Abdulrahman University, Riyadh 11671, Saudi Arabia; 4Department of Medical Laboratory Science, College of Applied Medical Sciences, University of Hail, Hail 55476, Saudi Arabia; 5Molecular Diagnostics and Personalized Therapeutics Unit, University of Hail, Hail 55476, Saudi Arabia; 6Department of Pharmacology and Toxicology, College of Pharmacy, University of Hail, Hail 55211, Saudi Arabia; 7Chemistry Department, Faculty of Science, Islamic University of Madinah, Madinah 42351, Saudi Arabia; 8Chemistry Department, Faculty of Science, Cairo University, Giza 12613, Egypt

**Keywords:** pyridines, 1,3,4-thiadiazoles, 1,3-thiazoles, hydrazonoyl halides, molecular docking, anticancer activity

## Abstract

Pyridine, 1,3,4-thiadiazole, and 1,3-thiazole derivatives have various biological activities, such as antimicrobial, analgesic, anticonvulsant, and antitubercular, as well as other anticipated biological properties, including anticancer activity. The starting 1-(3-cyano-4,6-dimethyl-2-oxopyridin-1(2*H*)-yl)-3-phenylthiourea (**2**) was prepared and reacted with various hydrazonoyl halides **3a**–**h,** α-haloketones **5a**–**d**, 3-chloropentane-2,4-dione **7a** and ethyl 2-chloro-3-oxobutanoate **7b**, which afforded the 3-aryl-5-substituted 1,3,4-thiadiazoles **4a**–**h,** 3-phenyl-4-arylthiazoles **6a**–**d** and the 4-methyl-3- phenyl-5-substituted thiazoles **8a,b**, respectively. The structures of the synthesized products were confirmed by spectral data. All of the compounds also showed remarkable anticancer activity against the cell line of human colon carcinoma (HTC-116) as well as hepatocellular carcinoma (HepG-2) compared with the Harmine as a reference under in vitro condition. 1,3,4-Thiadiazole **4h** was found to be most promising and an excellent performer against both cancer cell lines (IC_50_ = 2.03 ± 0.72 and 2.17 ± 0.83 µM, respectively), better than the reference drug (IC_50_ = 2.40 ± 0.12 and 2.54 ± 0.82 µM, respectively). In order to check the binding modes of the above thiadiazole derivatives, molecular docking studies were performed that established a binding site with EGFR TK.

## 1. Introduction

Designing new, effective, selective, highly potent, although more tolerant, anticancer drugs through the identification of novel structures remains a considerable challenge for the researchers in the field of medicinal chemistry. Hybrid drug design has emerged during the past few years as a leading technique for the creation of innovative anticancer medicines that, in theory, can address many of the pharmacokinetic drawbacks of conventional anticancer medications [1]. Medical researchers have focused on pyridines, 1,3,4-thiadiazole and 1,3-thiazole systems that have led to somewhat more effective and promising results in recent years. To name just a few, the following pyridine-based small compounds have received approval as anticancer medications: Vismodegib III, Crizotinib IV, Regorafenib II, and Sorafenib I (Figure 1) [2,3,4]. Different pyridine derivatives were studied for a variety of human cancer cell lines as a tool for novel anticancer drugs through the topoisomerase inhibitory activity. These results show various reports regarding different derivatives, such as bioisosteres of α-terthiophene as a potent for protein kinase C inhibitor [5], promising topoisomerase I and/or II inhibitory activity, as well as cytotoxicity against a variety of human cancer cell lines [6,7,8,9,10]. 

Among those, 1,3,4-thiadiazoles gained substantial interest due to their widespread biological activity, including antimicrobial, anti-inflammatory, antithrombotic, antihypertensive, antituberculosis, anesthetic, anticonvulsant and antiulcer activities [11,12,13,14,15,16]. Furthermore, different researchers also particularly report 1, 3, 4-thiadiazole derivatives for their excellent anticancer activity, which are confirmed by desirable IG_50_ and IC_50_ values in inhibitory effect, such as Filanesib and compounds I–III (Figure 1) [16,17,18,19,20].

1,3-Thiazoles, which derived from thiosemicarbazone derivatives, is also known for its various pharmacological applications, as its scaffold is useful for several natural, non-natural and semi-synthetic drugs, including anti-inflammatory, anti-parasitic and antineoplastic properties [13,21,22,23,24,25,26,27,28,29,30,31]. Numerous studies suggested that medications such as Tiazofurin, Dasatinib, and Dabrafenib that contain thiazoles may have anticancer properties against different cancer types (Figure 1) [32,33,34].

**Figure 1 molecules-27-06368-f001:**
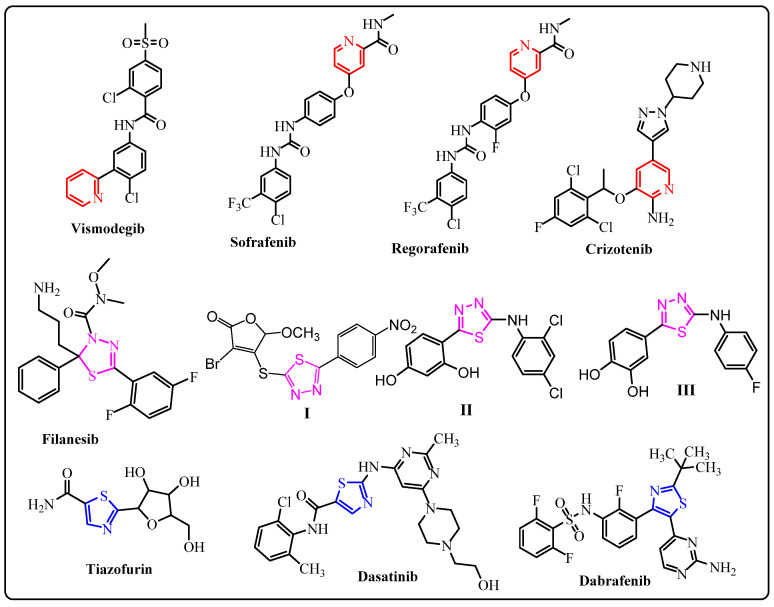
Examples of anticancer drugs bearing pyridine, thiazole, and thiadiazole moieties.

Under the influence of the findings mentioned above, continuous efforts were performed to synthesize innovative anticancer compounds [13,35,36,37,38,39,40,41,42,43,44,45,46]. The present report aims to elaborate on the new series of thiadiazole-pyridines as well as thiazole-pyridines that might have cytotoxic effects via the inhibition of protein Epidermal Growth Factor Tyrosine Kinase receptor (EGFR TK), which plays an essential mediating role in cell proliferation, angiogenesis, apoptosis, and metastatic spread compared with reference drugs.

## 2. Results and Discussion

1-(3-Cyano-4,6-dimethyl-2-oxopyridin-1(2*H*)-yl)-3-phenylthiourea (**2**) [47] was synthesized through the reaction of phenyl isothiocyanate with 1-amino-4,6-dimethyl-2-oxo-1,2-dihydropyridine-3-carbonitrile (**1**) under the influence of a catalytic amount of KOH in absolute ethanol as a solvent, as shown in Figure 1.

The conduct of compound **2** towards various hydrazonoyl chlorides was explored to synthesize a novel series of 1,3,4-thiadiazole derivatives. Therefore, when compound **2** was treated with hydrazonoyl chlorides **3a**–**h** in ethanol as a solvent in the presence of a TEA as a catalyst, it afforded a series of single product, recognized as 4,6-dimethyl-2-oxo-1-((3-aryl-5-substituted-1,3,4-thiadiazol-2(3*H*)-ylidene)amino)-1,2-dihydropyridine-3-carbonitriles **4a**–**h** (Figure 1). The 1,3,4-thiadiazole **4** was formed through the alkylation of the thiol group present in thiosemicarbazone moiety, an intramolecluar cyclization and finally elimination of aniline molecule.

All the structures of products **4a**–**h** were confirmed by elemental analyses followed by spectral data. In general, the ^1^H-NMR spectra of **4a** (see Appendix A), taken as an example**,** showed a singlet (1*H*) at *δ* 6.39 ppm corresponding to the pyridine proton, three singlets at *δ* 2.01, 2.33 and 2.46 ppm corresponding to the three CH_3_ groups and a multiplet *δ* 7.09–7.46 ppm corresponding to the five aromatic protons. The ^13^C-NMR spectrum of **4a** revealed two signals at *δ* = 25.6, 194.7 ppm, which is characteristic for the acetyl group (CH_3_C=O). The disappearance of the two NH absorption bands was also observed in the IR spectra because of the elimination of amine groups from the starting material **2**. Moreover, the mass spectrum of the products **4a**–**h** revealed a molecular ion peaks at the expected *m*/*z* values.

**Scheme 1 molecules-27-06368-sch001:**
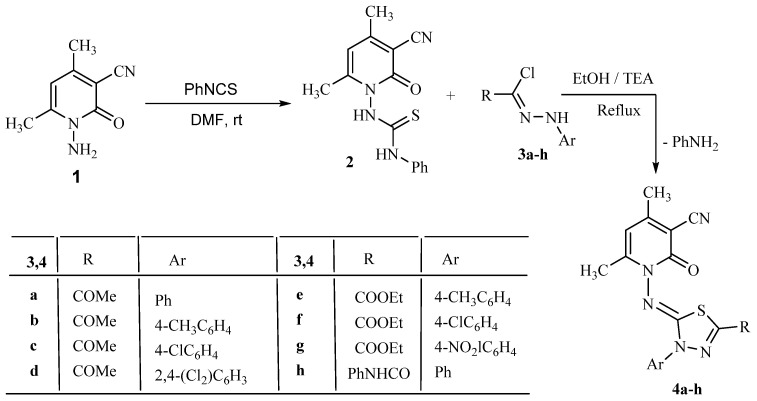
Synthesis of thiadiazoles **4a**–**h**.

In addition to this, the chemical reactivity of compound **2** with several of α-haloketones was also investigated to synthesize a series of novel thiazole derivatives. Accordingly, compound **2** reacted with α-haloketones **5a**–**d** in the presence of TEA as a catalyst under refluxing condition using EtOH as a solvent that resulted a corresponding thiazoles **6a**–**d** series, as shown in Figure 2 (see Experimental).

All the structures of the series of products **6a**–**e** were also confirmed through the analytical followed by spectral data analysis (see Experimental). Compound **6c** showed a typical singlet signal that appeared at *δ* 3.82, 6.55, and 6.80 ppm due to the OCH_3_ group, pyridine-H5, and thiazole-H5, respectively, in the ^1^H-NMR spectra. In addition to this, a multiplet is observed in the region: 7.01–7.47 ppm assignable to the nine aromatic hydrogens. On the other hand, the ^13^C-NMR spectrum of **6c** showed four signals at *δ* = 17.8, 21.2, 56.7 and 162.4 ppm characteristic for 2 Ar-CH_3_, Ar-OCH_3_ and C=O groups, respectively, in addition to sixteen aromatic carbon signals in the range of 107.3–149.4 ppm.

Finally, compound **2** reacted with α-chloro compounds **7a,b** under refluxing condition in the presence of EtOH as a solvent and TEA as a catalyst that resulted a single product, identified as 4,6-dimethyl-1-((4-methyl-3-phenyl-5-substitutedthiazol-2(3*H*)-ylidene) amino)4yridine-2(1*H*)-ones **8a,b**, as outlined in Figure 2.

The structure of the isolated product **8** was inferred from its IR and ^1^H-NMR spectral data and elemental analysis (see Experimental).

### 2.1. Anti-Cancer Activity

The series of prepared compounds **4a**–**h** and **6a**–**d** were investigated against human colon carcinoma (HCT-116) followed by the hepatocellular carcinoma (HepG2) cell lines to obtain pharmacological activities using Harmine as a reference drug through colorimetric MTT assay under in vitro conditions. The survival curve was obtained by plotting the relation between the concentrations of the drugs against the surviving cells, resulting in the 50% inhibitory concentration (IC_50_). The anti-proliferative activity is also achieved through the expression of the mean IC_50_ by three independent experiments (μM) ± standard deviation calculated from three replicates.

Table 1 and Figure 2 summarize the structure- and concentration-dependent anticancer activities of the series of compounds against HTC-116 cell lines. An in vitro inhibition activity shows a positive trend along with all tested compounds. Compounds such as **4c**, **4d**, **4f** and **4g** show comparable activity to that of Harmine (IC_50_ = 2.40 ± 0.12 µM) as a reference, whereas compound **4h** demonstrates even better results compared with the same reference. A similar trend of results was also observed for the hepatocellular carcinoma (HepG2) cell line assay, where **4c**, **4d**, **4f**, **4g** and **4h** show either comparable or improved inhibitory activity, with **6h** (IC_50_ = 2.17 ± 0.83 µM) showing the maximum effect in comparison with the reference Harmine (IC_50_ = 2.54 ± 0.82 µM). 

**Table 1 molecules-27-06368-t001:** The anticancer activity of the series of compounds **4a**–**h**, and **6a**–**d** towards human colon carcinoma (HCT-116) and hepatocellular carcinoma (HepG2) cell lines expressed as IC_50_ values (µM) ± standard deviation from three replicates.

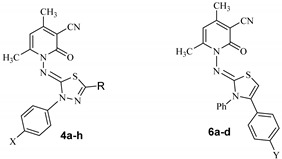				
Tested Compounds	R	X (or Y)	IC_50_ (µM)	
			HCT-116	HepG2
**4a**	COCH_3_	H	13.39 ± 1.04	16.44 ± 1.06
**4b**	COCH_3_	CH_3_	32.57 ± 2.37	37.56 ± 1.24
**4c**	COCH_3_	Cl	7.91 ± 0.83	9.18 ± 0.91
**4d**	COCH_3_	2,4-diCl	5.04 ± 0.59	7.32 ± 0.75
**4e**	COOEt	CH_3_	16.25 ± 1.05	19.35 ± 1.30
**4f**	COOEt	Cl	4.37 ± 0.28	6.94 ± 0.69
**4g**	COOEt	NO_2_	3.35 ± 0.46	3.94 ± 0.80
**4h**	CONHPh	H	2.03 ± 0.72	2.17 ± 0.83
**6a**	--	H	15.57 ± 1.30	19.12 ± 1.36
**6b**	--	CH_3_	36.29 ± 1.32	25.90 ± 0.70
**6c**	--	OCH_3_	21.00 ± 1.28	19.37 ± 1.29
**6d**	--	Cl	9.61 ± 0.88	7.36 ± 0.85
**Harmine**	--	--	2.40 ± 0.12	2.54 ± 0.82

For thiadiazoles **4a**–**h**: **4h** (amidophenyl, has a phenyl ring along with electron withdrawing amido group resulting strongest activity) > **4g** (strong electron withdrawing nitro group, increases activity) > **4f** (with ester group along with one electron withdrawing Cl atom) > **4d** (acetyl group with electron withdrawing 2 Cl atom) > **4c** (acetyl group with mild electron withdrawing one Cl atom) > **4a** (with acetyl group with un-substituted phenyl group) > **4e** (ester with methyl group) > **4b** (acetyl group with methyl group, electron donating group decreases activity). Overall, electron releasing groups decrease the activity, whereas strong electron withdrawing groups increase the activity. A selective high activity is observed, particularly with **4h**, possibly due to the fact that **4h** possess one extra phenyl ring connected with the pyridine group in the amido side, which significantly enhances its aromatic π-π interaction with the Phe, Tyr and Trp residues. This is in coherence with the harmine where one phenyl and one pyridine is fused with an indole group, resulting in the same kind of interaction with amino acid residues containing an aromatic group. In addition to this, a noteworthy mention would be, in **4h**, the nitrogen in the amido group is engaged in a tautomeric structure, thereby restricting the electron releasing power of nitrogen in the phenyl ring. Such interactions are absent for the rest of thiadiazole derivatives **4a**–**g**, where only electron withdrawing group is present in the lone available phenyl ring. 

This is in analogous with the thiazoles derivatives **6a**–**d**, where only electron releasing groups are present with the single phenyl group available, resulting in less activity compared to thiadiazoles **4a**–**h** with the only exception of **6d** (an electron withdrawing Cl atom is present), where moderate anticancer activity is detected (Figure 3).

### 2.2. Docking Study for Cytotoxicity 

The protein Epidermal Growth Factor Receptor Tyrosine Kinase Domain (EGFR TK) was selected for this study, where the ability to inhibit this receptor ultimately leads to the blockade of the growth pathway, giving a promising anti-cancer agent [48]. The lower binding energy resulting from the association of the compound with the targeted protein is an indication of a higher binding efficiency. The results of the docking protocol were validated by the re-docking of the co-crystallized ligand (W19) inside the active site of EGFR TK (Figure 4 and Figure 5). Harmine was used in this study as an EGFR TK inhibitor. By comparing the binding affinity of different screened synthesized compounds with Harmine (ΔG of –7.1), it was found that compound **4h** showed the best binding affinity with ΔG of −10.8, and **4b**, **4c**, **4d**, **4e**, **4f**, **6a**, **6b**, and **6c** showed binding activity ΔG −8.1 to −9.2. The screened compounds showed a possible interaction with EGFR TK active sites as depicted in Table 2 and Figure 6, Figure 7, Figure 8, Figure 9, Figure 10, Figure 11, Figure 12 and Figure 13.

**Table 2 molecules-27-06368-t002:** The binding scores and interactions of the examined compounds and the Harmine inhibitor inside the binding pocket of receptor of (3W33) for EGFR TK.

Compounds	Binding Scores (kcal/mol)	Hydrogen Bond Interactions	Distance (Å)	Hydrophobic Interactions	Distance (Å)
**4a**	−8.5	MET769	2.08	LEU694 VAL702 LYS721 LEU820 THR830	3.78 3.75 3.46, 3.93 3.24, 3.90 3.77
**4b**	−9.2	MET769	1.96	LEU694 VAL702 ALA719 LYS721 LEU764 LEU820	3.74 3.69 3.89 3.48, 3.71 3.77 3.39
**4c**	−8.9	MET769	2.07	LEU694 VAL702 ALA719 LYS721 LEU820 THR830	3.78 3.69 3.87 3.47 3.21, 3.94 3.75
**4d**	−9.2	GLU738 THR830 ASP831 PHE832	2.71 2.64 2.20 3.17	LEU694 PHE699 VAL702 ALA719 LYS721 LEU764 THR766 LEU820	3.55 3.66 3.04 3.56 3.95 3.59 3.50 3.58
**4e**	−8.7	GLU738 THR830 ASP831 PHE832	2.71 2.64 2.20 3.17	LEU694 PHE699 VAL702 ALA719 LYS721 LEU764 THR766 LEU820	3.55 3.66 3.04 3.56 3.95 3.59 3.50 3.58
**4f**	−8.8	MET769	2.10	LEU694 VAL702 ALA719 LYS721 LEU820 THR830	3.88 3.58 3.87 3.42 3.33, 3.86 3.92
**4g**	−9.1	MET769	2.56	LEU694 VAL702 ALA719 LYS721 LEU820	3.74 3.38 3.68 3.59 3.31, 3.81
**4h**	−10.8	ARG841 ASN842 LYS745 THR854	2.08 2.27 3.48 3.45	LEU718 VAL726 ALA743 LYS745 LEU788 THR790	3.74, 3.50 3.61, 3.30 3.67 3.36, 3.89 3.66 3.80
**6a**	−8.8			LEU694 VAL702 ALA719 LYS721 LEU820	3.69 3.13, 3.91 3.61 3.73, 3.87 3.41, 3.64
**6b**	−8.9	CYS797	2.29	LEU718 VAL726 ASP855 LEU777 LEU788 THR790 LEU844 PHE997	3.90, 3.39, 3.47 3.64 3.67 3.08 3.48 3.48 3.53 3.29
**6c**	−7.9	THR766	2.38	LEU694 VAL702 LYS721 LEU820	3.39 3.32, 3.51 3.69, 3.73 3.39
**6d**	−8.8	THR766	2.58	PHE699 VAL702 ALA719 MET769 ARG817 LEU820	3.57 3.95 3.50 3.93 3.78 3.91, 3.48
**Harmine**	−7.1			LEU718 VAL726 LYS745 THR790 LEU792	3.61 3.58, 3.64 3.76 3.63 3.72
**W19**	−10.8	LYS745	2.36	LEU718 VAL726 LYS745 LEU777 LEU788 THR790 THR845	3.67 3.65, 3.94 3.85 3.79 3.97 3.73 3.79

## 3. Experimental

Elementar vario LIII CHNS analyzer (Elementar Analysensysteme GmbH, Langenselbold, Germany) is used to measure all elemental analysis. Electrothermal IA 9000 series Digital Melting Point Apparatus (Shanghai Jiahang Instruments Co., Jiading District, Shanghai, China) was used to obtain melting points data. Shimadzu FTIR 8101 PC infrared spectrophotometers (Shimadzu Co., Kyoto, Japan) were used to record IR spectra data in KBr discs on Pye Unicam SP 3300. Varian Mercury VX-300 NMR spectrometer (Bruker Biospin, Karlsruhe, Germany) was used with the operating frequency of 300 MHz (^1^H-NMR) in deuterated dimethylsulfoxide (DMSO-*d*6) solvent to record NMR spectra, where chemical shifts were related to the solvent used. Shimadzu GCeMS-QP1000 EX mass spectrometer (Shimadzu Co., Kyoto, Japan) was used to record mass spectra at 70 eV. The cytotoxicity of the prepared compounds was measured by the Regional Center for Mycology and Biotechnology in Al-Azhar University, Cairo, Egypt. 


**Synthesis of 1-(3-cyano-4,6-dimethyl-2-oxopyridin-1(2*H*)-yl)-3-phenylthiourea (2)**


A mixture of 1-amino-4,6-dimethyl-2-oxo-1,2-dihydropyridine-3-carbonitrile (**1**) (1.63 g, 10 mmol), KOH (0.56 g, 10 mmol) in DMF (30 mL), was stirred for 10 min. Then, PhNCS (1.35 g, 10 mmol) is added under stirring condition and continued for the next 6 h. Afterwards, the solution was diluted with 30 mL of distilled water, followed by neutralization by adding aqueous AcOH dropwise, resulting in a solid recrystallized from dioxin to obtain yellowish brown crystals (71%) as a pure product of compound **2**; mp = 209–211 °C (Lit mp = 205–207 °C [47]); ^1^H-NMR (DMSO-*d*_6_): *δ* 2.22 (s, 3H, CH_3_), 2.30 (s, 3H, CH_3_), 6.34 (s, 1H, Pyridine-H5), 7.32–7.69 (m, 5H, Ar-H), 8.55 (s, br, 1H, NH), 8.95 (s, br, 1H, NH) ppm; IR (KBr): *v* 3372, 3241 (2NH), 3033, 2951 (CH), 2218 (CN), 1675 (C=O), 1599 (C=N), 1335 (C=S) cm^−1^; MS *m*/*z* (%): 298 (M^+^, 85). Anal Calcd for C_15_H_14_N_4_OS (298.36): C, 60.38; H, 4.73; N, 18.78. Found: C, 60.24; H, 4.59; N, 18.58%.


**Synthesis of 4,6-dimethyl-2-oxo-1-((3-aryl-5-substituted-1,3,4-thiadiazol-2(3*H*)-ylidene)amino)-1,2-dihydropyridine-3-carbonitriles (4a–h).**


A mixture of compound **2** (0.298 g, 1 mmol) and appropriate hydrazonoyl halides **3a**–**h** (1 mmol) in DMF (20 mL) containing Et_3_N (0.1 g, 1 mmol) was heated under reflux for 3–6 h. The resultant solid product was recrystallized by appropriate solvent to give thiadiazoles **4a**–**h**. Below is a list of the spectrum information and physical characteristics of the products **4a**–**h**.


**1-((5-Acetyl-3-phenyl-1,3,4-thiadiazol-2(3*H*)-ylidene)amino)-4,6-dimethyl-2-oxo-1,2-dihydropyridine-3-carbonitrile (4a).**


Yellow solid (79%); m.p. 233–235 °C (DMF); ^1^H-NMR (DMSO-*d*_6_): *δ* 2.01 (s, 3H, CH_3_), 2.33 (s, 3H, CH_3_), 2.46 (s, 3H, CH_3_), 6.39 (s, 1H, Pyridine-H5), 7.09–7.46 (m, 5H, Ar-H) ppm; ^13^C-NMR (DMSO-*d*_6_): *δ* 19.6, 21.2, 25.6 (3CH_3_), 107.8, 115.9, 116.4, 118.6, 122.7, 123.1, 124.0, 125.8, 138.1, 142.3, 152.3 (Ar-C and C=N), 163.2, 194.7 (2 C=O) ppm; IR (KBr): *v* 3047, 2933 (CH), 2220 (CN),1704, 1651 (C=O), 1599 (C=N) cm^−1^; MS *m*/*z* (%): 365 (M^+^, 49). Anal. Calcd. for C_18_H_15_N_5_O_2_S (365.41): C, 59.17; H, 4.14; N, 19.17. Found C, 59.30; H, 4.04; N, 19.11%.


**1-((5-Acetyl-3-(*p*-tolyl)-1,3,4-thiadiazol-2(3*H*)-ylidene)amino)-4,6-dimethyl-2-oxo-1,2-dihydropyridine-3-carbonitrile (4b).**


Yellow solid (80%); m.p. 243–245 °C (DMF); ^1^H-NMR (DMSO-*d*_6_): *δ* 2.01 (s, 3H, CH_3_), 2.32 (s, 3H, CH_3_), 2.40 (s, 3H, CH_3_), 2.45 (s, 3H, CH_3_), 6.39 (s, 1H, Pyridine-H5), 7.02–7.80 (m, 4H, Ar-H) ppm; ^13^C-NMR (DMSO-*d*_6_): *δ* 19.1, 20.7, 21.2, 25.4 (4CH_3_), 107.3, 116.5, 122.9, 127.5, 130.1, 130.6, 133.8, 138.2, 145.2, 154.2 (Ar-C and C=N), 162.1, 194.2 (2 C=O) ppm; IR (KBr): *v* 3028, 2940 (CH), 2217 (CN),1703, 1667 (C=O), 1597 (C=N) cm^−1^; MS *m*/*z* (%): 379 (M^+^, 38). Anal. Calcd. for C_19_H_17_N_5_O_2_S (379.44): C, 60.14; H, 4.52; N, 18.46. Found C, 60.05; H, 4.42; N, 18.29%.


**1-((5-Acetyl-3-(4-chlorophenyl)-1,3,4-thiadiazol-2(3*H*)-ylidene)amino)-4,6-dimethyl-2-oxo-1,2-dihydropyridine-3-carbonitrile (4c).**


Yellow solid (78%); m.p. 228–230 °C (dioxane); ^1^H-NMR (DMSO-*d*_6_): *δ* 2.03 (s, 3H, CH_3_), 2.25 (s, 3H, CH_3_), 2.41 (s, 3H, CH_3_), 6.39 (s, 1H, Pyridine-H5), 7.11–7.85 (m, 4H, Ar-H) ppm; IR (KBr): *v* 3052, 2944 (CH), 2218 (CN), 1709, 1663 (C=O), 1598 (C=N) cm^−1^; MS *m*/*z* (%): 401 (M^+^_+_ 2, 31), 399 (M^+^, 100). Anal. Calcd. for C_18_H_14_ClN_5_O_2_S (399.85): C, 54.07; H, 3.53; N, 17.52. Found C, 54.01; H, 3.36; N, 17.48%.


**1-((5-Acetyl-3-(2,4-dichlorophenyl)-1,3,4-thiadiazol-2(3*H*)-ylidene)amino)-4,6-dimethyl-2-oxo-1,2-dihydropyridine-3-carbonitrile (4d).**


Brown solid (79%); m.p. 262–264 °C (DMF); ^1^H-NMR (DMSO-*d*_6_): *δ* 2.04 (s, 3H, CH_3_), 2.31 (s, 3H, CH_3_), 2.43 (s, 3H, CH_3_), 6.39 (s, 1H, Pyridine-H5), 7.26–7.86 (m, 3H, Ar-H) ppm; IR (KBr): *v* 3047, 2937 (CH), 2222 (CN), 1711, 1672 (C=O), 1601 (C=N) cm^−1^; MS *m*/*z* (%): 434 (M^+^, 81). Anal. Calcd. for C_18_H_13_Cl_2_N_5_O_2_S (434.30): C, 49.78; H, 3.02; N, 16.13. Found C, 49.83; H, 3.00; N, 16.04%.


**Ethyl 5-((3-cyano-4,6-dimethyl-2-oxopyridin-1(2*H*)-yl)imino)-4-(p-tolyl)-4,5-dihydro-1,3,4-thiadiazole-2-carboxylate (4e).**


Yellow solid (77%); m.p. 189–191 °C (EtOH\DMF); ^1^H-NMR (DMSO-*d*_6_): *δ* 1.17–1.21 (t, 3H, CH_3_), 2.01 (s, 3H, CH_3_), 2.32 (s, 3H, CH_3_), 2.42 (s, 3H, CH_3_), 4.13–4.17 (q, 2H, CH_2_), 6.45 (s, 1H, Pyridine-H5), 7.12–7.59 (m, 4H, Ar-H) ppm; ^13^C-NMR (DMSO-*d*_6_): *δ* 12.9, 17.5, 21.1, 21.7 (4CH_3_), 117.3, 119.0, 121.2, 121.7, 122.4, 124.1, 125.0, 130.1, 139.4, 142.5, 151.2 (Ar-C and C=N), 161.2, 163.5 (2 C=O) ppm; IR (KBr): *v* 3049, 2930 (CH), 2219 (CN), 1723, 1669 (C=O), 1600 (C=N) cm^−1^; MS *m*/*z* (%): 409 (M^+^, 14). Anal. Calcd. for C_20_H_19_N_5_O_3_S (409.46): C, 58.67; H, 4.68; N, 17.10. Found C, 58.52; H, 4.55; N, 17.02%.


**Ethyl 4-(4-chlorophenyl)-5-((3-cyano-4,6-dimethyl-2-oxopyridin-1(2*H*)-yl)imino)-4,5-dihydro-1,3,4-thiadiazole-2-carboxylate (4f).**


Yellow solid (77%); m.p. 185–187 °C (EtOH); ^1^H-NMR (DMSO-*d*_6_): *δ* 1.20–1.27 (t, 3H, CH_3_), 2.03 (s, 3H, CH_3_), 2.38 (s, 3H, CH_3_), 4.11–4.17 (q, 2H, CH_2_), 6.46 (s, 1H, Pyridine-H5), 7.23–7.60 (m, 4H, Ar-H) ppm; IR (KBr): *v* 3046, 2937 (CH), 2217 (CN),1722, 1660 (C=O), 1601 (C=N) cm^−1^; MS *m*/*z* (%): 431 (M^+^_+_ 2, 20), 429 (M^+^, 63). Anal. Calcd. for C_19_H_16_ClN_5_O_3_S (429.88): C, 53.09; H, 3.75; N, 16.29. Found C, 53.18; H, 3.58; N, 16.14%.


**Ethyl 5-((3-cyano-4,6-dimethyl-2-oxopyridin-1(2*H*)-yl)imino)-4-(4-nitrophenyl)-4,5-dihydro-1,3,4-thiadiazole-2-carboxylate (4g).**


Yellow solid (73%); mp = 217–219 °C (EtOH\DMF); ^1^H-NMR (DMSO-*d*_6_): *δ* 1.20–1.27 (t, 3H, CH_3_), 2.29 (s, 3H, CH_3_), 2.42 (s, 3H, CH_3_), 4.18–4.26 (q, 2H, CH_2_), 6.84 (s, 1H, Pyridine-H5), 7.27–8.34 (m, 4H, Ar-H) ppm; IR (KBr): *v* 3039, 2943 (CH), 2218 (CN),1719, 1665 (C=O), 1600 (C=N) cm^−1^; MS *m*/*z* (%): 440 (M^+^, 70). Anal. Calcd. for C_19_H_16_N_6_O_5_S (440.43): C, 51.81; H, 3.66; N, 19.08. Found C, 51.66; H, 3.50; N, 19.03%.


**5-((3-Cyano-4,6-dimethyl-2-oxopyridin-1(2*H*)-yl)imino)-N,4-diphenyl-4,5-dihydro-1,3,4-thiadiazole-2-carboxamide (4h).**


Yellow solid (86%); mp = 277–279 °C (DMF); ^1^H-NMR (DMSO-*d*_6_): *δ* 2.16 (s, 3H, CH_3_), 2.38 (s, 3H, CH_3_), 6.42 (s, 1H, Pyridine-H5), 7.03–7.79 (m, 10H, Ar-H), 10.21 (s, 1H, NH) ppm; ^13^C-NMR (DMSO-*d*_6_): *δ* 17.9, 21.4 (2CH_3_), 112.4, 115.9, 119.3, 120.5, 121.5, 122.1, 124.3, 124.7, 125.2, 128.6, 132.4, 134.6, 139.4, 141.5, 152.4 (Ar-C and C=N), 161.2, 162.7 (2 C=O) ppm; IR (KBr): *v* 3278 (NH), 3061, 2947 (CH), 2219 (CN), 1678, 1663 (C=O), 1597 (C=N) cm^−1^; MS *m*/*z* (%): 442 (M^+^, 18). Anal. Calcd. for C_23_H_18_N_6_O_2_S (442.50): C, 62.43; H, 4.10; N, 18.99. Found C, 62.52; H, 4.04; N, 18.75%.


**Synthesis of thiazoles 6a–d or 8a,b.**


A mixture of **2** (0.298 g, 1 mmol) and *α*-haloketones **5a**–**d** or 3-chloropentane-2,4-dione (**7a**) or ethyl 2-chloro-3-oxobutanoate (**7b**) (1 mmol) in DMF (15 mL) was refluxed for 4–6 h and was continuously monitored in TLC. The separation of the product was clearly observed during the course of the reaction. The resultant solid product was then filtered, washed several times with water, dried and recrystallized in the proper solvent to give the corresponding thiazoles **6a**–**d** or **8a,b**, respectively.


**1-((3,4-Diphenylthiazol-2(3*H*)-ylidene)amino)-4,6-dimethyl-2-oxo-1,2-dihydropyridine-3-carbonitrile (6a).**


Yellow solid (74%); mp = 230–232 °C (DMF); ^1^H-NMR (DMSO-*d*_6_): *δ* 2.23 (s, 3H, CH_3_), 2.34 (s, 3H, CH_3_), 6.37 (s, 1H, Pyridine-H5), 6.67 (s, 1H, Thiazole-H5), 7.19–8.05 (m, 10H, Ar-H) ppm; IR (KBr): *v* 3047, 2934 (CH), 2217 (CN), 1667 (C=O), 1599 (C=N) cm^−1^; MS *m*/*z* (%): 398 (M^+^, 37). Anal. Calcd. for C_23_H_18_N_4_OS (398.48): C, 69.33; H, 4.55; N, 14.06. Found C, 69.17; H, 4.42; N, 14.04%.


**4,6-Dimethyl-2-oxo-1-((3-phenyl-4-(*p*-tolyl)thiazol-2(3*H*)-ylidene)amino)-1,2-dihydropyridine-3-carbonitrile (6b).**


Yellow solid (77%); mp = 213–215 °C (EtOH); ^1^H-NMR (DMSO-*d*_6_): *δ* 2.23 (s, 3H, CH_3_), 2.24 (s, 3H, CH_3_), 2.34 (s, 3H, CH_3_), 6.37 (s, 1H, Pyridine-H5), 7.07 (s, 1H, Thiazole-H5), 7.33–7.44 (m, 9H, Ar-H) ppm; IR (KBr): *v* 3042, 2950 (CH), 2219 (CN), 1671 (C=O), 1602 (C=N) cm^−1^; MS *m*/*z* (%): 412 (M^+^, 18). Anal. Calcd. for C_24_H_20_N_4_OS (412.51): C, 69.88; H, 4.89; N, 13.58. Found C, 69.70; H, 4.83; N, 13.37%.


**1-((4-(4-Methoxyphenyl)-3-phenylthiazol-2(3*H*)-ylidene)amino)-4,6-dimethyl-2-oxo-1,2-dihydropyridine-3-carbonitrile (6c).**


Yellow solid (80%); mp = 221–223 °C (EtOH\DMF); ^1^H-NMR (DMSO-*d*_6_): *δ* 2.23 (s, 3H, CH_3_), 2.32 (s, 3H, CH_3_), 3.82 (s, 3H, OCH_3_), 6.55 (s, 1H, Pyridine-H5), 6.80 (s, 1H, Thiazole-H5), 7.01–7.47 (m, 9H, Ar-H) ppm; ^13^C-NMR (DMSO-*d*_6_): *δ* 17.8, 21.2 (2CH_3_), 56.7 (OCH_3_), 107.8, 113.9, 115.0, 118.3, 121.9, 122.5, 123.6, 130.0, 131.8, 135.6, 138.9, 140.8, 144.0, 152.0, 152.5, 154.3 (Ar-C and C=N), 162.4 (C=O) ppm; IR (KBr): *v* 3074, 2928 (CH), 2218 (CN), 1683 (C=O), 1603 (C=N) cm^−1^; MS *m*/*z* (%): 403 (M^+^, 23). Anal. Calcd. for C_24_H_20_N_4_O_2_S (428.51): C, 67.27; H, 4.70; N, 13.08. Found C, 67.36; H, 4.61; N, 13.02%.


**1-((4-(4-Chlorophenyl)-3-phenylthiazol-2(3*H*)-ylidene)amino)-4,6-dimethyl-2-oxo-1,2-dihydropyridine-3-carbonitrile (6d).**


Yellow solid (79%); mp = 217–219 °C (DMF); ^1^H-NMR (DMSO-*d*_6_): *δ* 2.23 (s, 3H, CH_3_), 2.34 (s, 3H, CH_3_), 6.76 (s, 1H, Pyridine-H5), 7.01 (s, 1H, Thiazole-H5), 7.12–7.77 (m, 9H, Ar-H) ppm;IR (KBr): *v* 3048, 2927 (CH), 2218 (CN), 1669 (C=O), 1602 (C=N) cm^−1^; MS *m*/*z* (%): 409 (M^+^_+_ 2, 13), 407 (M^+^, 41). Anal. Calcd. for C_23_H_17_ClN_4_OS (432.93): C, 63.81; H, 3.96; N, 12.94. Found C, 63.62; H, 3.77; N, 12.73%.


**1-((5-Acetyl-4-methyl-3-phenylthiazol-2(3*H*)-ylidene)amino)-4,6-dimethyl-2-oxo-1,2-dihydropyridine-3-carbonitrile (8a).**


Yellow solid (83%); mp = 195–197 °C (EtOH); ^1^H-NMR (DMSO-*d*_6_): *δ* 2.18 (s, 3H, CH_3_), 2.25 (s, 3H, CH_3_), 2.33 (s, 3H, CH_3_), 2.43 (s, 3H, CH_3_), 6.36 (s, 1H, Pyridine-H5), 7.55–7.66 (m, 5H, Ar-H) ppm; IR (KBr): *v* 3048, 2935 (CH), 2219 (CN),1709, 1671 (C=O), 1601 (C=N) cm^−1^; MS *m*/*z* (%): 353 (M^+^, 35). Anal. Calcd. for C_20_H_18_N_4_O_2_S (378.45): C, 63.47; H, 4.79; N, 14.80. Found C, 63.35; H, 4.62; N, 14.69%.


**Ethyl 2-((3-cyano-4,6-dimethyl-2-oxopyridin-1(2*H*)-yl)imino)-4-methyl-3-phenyl-2,3-dihydrothiazole-5-carboxylate (8b).**


Yellow solid (85); mp = 190–192 °C (EtOH); ^1^H-NMR (DMSO-*d*_6_): *δ* 1.21–1.25 (t, 3H, CH_3_), 2.19 (s, 3H, CH_3_), 2.22 (s, 3H, CH_3_), 2.38 (s, 3H, CH_3_), 4.21–4.25 (q, 2H, CH_2_), 6.39 (s, 1H, Pyridine-H5), 7.45–7.62 (m, 5H, Ar-H) ppm; ^13^C-NMR (DMSO-*d*_6_): *δ* 14.5, 17.1, 21.3 (3CH_3_), 62.5 (CH_2_), 106.4, 113.3, 115.3, 119.8, 123.7, 124.5, 127.6, 131.5, 133.0, 140.2, 144.1, 148.7 (Ar-C and C=N), 162.5, 164.1 (2 C=O) ppm; IR (KBr): *v* 3037, 2943 (CH), 2218 (CN), 1724, 1668 (C=O), 1600 (C=N) cm^−1^; MS *m*/*z* (%): 408 (M^+^, 74). Anal. Calcd. for C_21_H_20_N_4_O_3_S (408.48): C, 61.75; H, 4.94; N, 13.72. Found C, 61.63; H, 4.81; N, 13.53%.

### 3.1. Cytotoxic Activity

The cytotoxicity of the newly synthesized series of compounds was studied against HCT-116 and HepG2 cells using the MTT assay through an incubation period of 24 h [49,50].

Mammalian cell line: HCT-116 and HepG2 cells were collected from VACSERA Tissue Culture Unit, Cairo, Egypt.

### 3.2. Docking Method

The MOE 2019.012 suite (Chemical Computing Group ULC, Montreal, Canada) [51] was applied in order to carry all docking studies for the newly synthesized compounds to suggest their plausible mechanism of action as the protein Epidermal Growth Factor Receptor Tyrosine Kinase Domain (EGFR TK) inhibitors by evaluating their binding grooves and modes to compare with Harmine as a reference standard. 

The newly prepared derivatives were placed into the MOE window, where they were treated to partial charge addition and energy minimization [52,53]. The produced compounds also were placed into a single database with the Harmine and saved as an MDB file, which was then uploaded to the ligand icon during the docking process. The Protein Data Bank was used to generate an X-ray of the targeted Epidermal Growth Factor Receptor Tyrosine Kinase Domain (EGFR TK) 3W33. Available online: https://www.rcsb.org/structure/3W33 (accessed on 17 July 2022). [48]. Furthermore, it was readied for the docking process by following the previously detailed stages [54,55]. Additionally, the downloaded protein was error-corrected, 3D hydrogen-loaded and energy-minimized [56,57].

In a general docking procedure, the newly prepared derivatives were substituted for the ligand site. After modifying the default program requirements previously stated, the co-crystallized ligand site was selected as the docking site, and the docking process was started [58]. In a nutshell, the docking site was chosen using the dummy atoms method [59]. The placement and scoring procedures, respectively, Triangle matcher and London dG, were chosen. The stiff receptor was used as the scoring method, and the GBVI/WSA dG was used as the refining method, respectively, to select the top 10 poses out of a total of 100 poses for each docked molecule [60,61]. For further research, the optimal pose for each ligand with the highest score, binding mode and RMSD value was chosen. It is important to note that the applied MOE program underwent the first step of program verification by docking Harmine to its ligand binding of the prepared Target [62,63]. By obtaining a low RMSD value (1.43) between the newly created compounds with docked Harmine, a valid performance was demonstrated.

## 4. Conclusions

In this paper, two new series of aryl substituted novel pyridine-1,3,4-thiadiazoles, and pyridine-thiazoles were synthesized starting with 1-(3-cyano-4,6-dimethyl-2-oxopyridin-1(2*H*)-yl)-3-phenylthiourea and several available reagents. All the structures were confirmed through elemental and spectral analysis, where the plausible mechanistic approach for their formation was discussed. All the prepared derivatives showed effectiveness towards the inhibition of human colon carcinoma (HCT-116) as well as hepatocellular carcinoma (HepG2) cell lines through in vitro evaluation and an in silico docking study. From the obtained results, compound **4h** (amidophenyl has a phenyl ring along with the electron withdrawing amido group resulting in the strongest activity) was found to be the strongest and most effective with 2.03 ± 0.72 μM against HCT-116, contributing its activity through a variety of interactions, such as hydrophobic interaction, hydrogen bonding in addition to aromatic stacking interactions with selected target (EGFR TK) pockets, compared with Harmine as a reference drug. A detailed analysis of all the series confirmed the electron withdrawing group present in the aryl substitution, resulting in the enhancement of anticancer activity, which could be promising for the future generation of new efficient anticancer drugs based on 1,3,5-thiadiazole and 1,3-thiazole derivatives.

## Data Availability

The data presented in this study are available on request from corresponding author.

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
