# Peer review of "Synthesis, Molecular Docking Study, and Cytotoxicity Evaluation of Some Novel 1,3,4-Thiadiazole as Well as 1,3-Thiazole Derivatives Bearing a Pyridine Moiety"

_molecules, 2022, doi:10.3390/molecules27196368_

Round 1

Reviewer 1 Report

The  article titled  Synthesis, Molecular Docking Study, and Cytotoxicity Evalua-2 tion of Some Novel 1,3,4-Thiadiazole as well as 1,3-Thiazole 3 Derivatives Bearing a Pyridine Moiety of the following  comments.

.

1)      Abstract, 2 should be between brackets

2)      Abstract, add number to all compounds

3)      Abstract , add result of compound 6h and standard.

4)      rational of this study should be improved depending on SAR of reported anticancer  drugs or compounds

5)      Abbreviations mean should be mention where firstly appeared

6)      Result and discussion, some compounds have more than three methyl groups (6 b ,e) line 90 so  discussion should be had more details.

7)      schemes have structures between brackets but this structure were not confirmed or separated so it is better to remove

8)      it is better to make CNMR for all compounds.

9)      Also discussion of CNMR should be added.

10)   At conclusion, authors  should mention why 6h is highly active why other compounds  are not so conclude SAR for their work

Author Response

Reviewer 1

Dear reviewer! Many thanks for your time spending and efforts to review the manuscript.

  • Abstract, 2 should be between brackets

Response: Done

  • Abstract, add number to all compounds

Response: Done (see revised manuscript)

  • Abstract, add result of compound 4h and standard.

Response: Done

  • Rational of this study should be improved depending on SAR of reported anticancer drugs or compounds

Response: the introduction was updated to include reported anticancer drugs bearing pyridine, thiazole, and thiadiazole moieties  (see revised manuscript)

  • Abbreviations mean should be mention where firstly appeared

Response: Done

  • Result and discussion, some compounds have more than three methyl groups (4b ,e) line 90 so  discussion should be had more details.

Response: The cited spectral data assigned to compound 4a

  • Schemes have structures between brackets but this structure were not confirmed or separated so it is better to remove

Response: Intermediates in schemes 1& 2 were removed and numbers updated

  • it is better to make CNMR for all compounds.

Response: We measured 13C-NMR for extra 4 compounds (see revised manuscript)

Compound 4b: 13C-NMR (DMSO-d6): δ 19.1, 20.6, 21.3, 25.3 (4CH3), 107.1, 116.4, 116.9, 119.5, 121.4, 122.7, 125.0, 125.6, 132.4, 140.6, 151.7 (Ar-C and C=N), 162.6, 194.4 (2 C=O) ppm;

Compound 4e: 13C-NMR (DMSO-d6): δ 12.9, 17.5, 21.1, 21.7 (4CH3), 117.3, 119.0, 121.2, 121.7, 122.4, 124.1, 125.0, 130.1, 139.4, 142.5, 151.2 (Ar-C and C=N), 161.2, 163.5 (2 C=O) ppm;

Compound 4h:13C-NMR (DMSO-d6): δ 17.9, 21.4 (2CH3), 112.4, 115.9, 119.3, 120.5, 121.5, 122.1, 124.3, 124.7, 125.2, 128.6, 132.4, 134.6, 139.4, 141.5, 152.4 (Ar-C and C=N), 161.2, 162.7 (2 C=O) ppm;

Compound 6c: 13C-NMR (DMSO-d6): δ 17.3, 21.1 (2CH3), 56.5 (OCH3), 107.3, 114.0, 116.2, 116.9, 121.5, 123.1, 124.4, 124.7, 127.6, 130.0, 131.4, 133.7, 138.6, 140.1, 144.6, 149.4 (Ar-C and C=N), 162.2 (C=O) ppm;

  • Also discussion of CNMR should be added.

Response: Done (see revised manuscript)

  • At conclusion, authors should mention why 4h is highly active why other compounds  are not so conclude SAR for their work

Response: Activity of compound 4h is due to presence of amidophenyl group, has a phenyl ring along with electron withdrawing amido group resulting strongest activity

Reviewer 2 Report

The author proposed the novel 1,3-Thiazole Derivatives as anticancer compounds using in silico and experimental validation (anticancer activity only). There are ample amounts of work done by researchers on this topic. The researchers had done various activities such as  Antibacterial and Antifungal activity, Anticancer activity, Anti-Helicobacter pylori, Anticonvulsant activity, Anti-inflammatory activity (COX-inhibitors), Antitubercular activity, Antiviral activity, Anti-leishmanial activity, Carbonic anhydrase inhibitory activity. What is the novelty of these compounds? The docking studies are also not competitive. There is some major concern about the study, please justify them.

1.  Why did the authors have chosen the 1M17 PDB entry? 

2. There are many PDB entries available in the PDB in recent years. Have the authors reviewed them?

3. The PDB entry 1m17 contains one native ligand (AQ4). Did the authors perform a re-docking of it?

4. Please provide the docking software details with the license and other details.

5. Molecular docking is not enough nowadays. It requires Molecular dynamics simulations. Why did the authors have not done MD simulation studies?

6. The docking image quality is very poor and not understandable.

Author Response

Dear reviewer! Many thanks for your time spending and efforts to review the manuscript.

What is the novelty of these compounds?

Response: it was reported that pyridine, 1,3,4-thiadiazole, and 1,3-thiazole derivatives have  anticancer properties. Under the influence of the findings, our continuous efforts were performed to synthesize innovative anticancer compounds based on thiazole and thiadiazole derivatives [35-46]. The present report aims to elaborate about the new series of thiazole-pyridines as well as thiadiazole-pyridines that might have cytotoxic effects via inhibition of protein Epidermal Growth Factor Tyrosine Kinase receptor (EGFR TK) which plays an essential mediating role in cell proliferation, angiogenesis, apoptosis, and met-astatic spread compared with reference drugs. All of the synthesized compounds also showed remarkable anticancer activity against the Cell line of human colon carcinoma (HTC-116) as well as hepatocellular carcinoma (HepG-2) compared with the Harmine as a reference under in-vitro condition. Compound like 1,3,4-thiadiazole 4h found to be most promising and an excellent performer against both cancer cell lines (IC50 = 2.03 ± 0.72 and 2.17 ± 0.83 μM, respectively) bettered than the reference drug (IC50 = 2.40 ± 0.12 and 2.54 ± 0.82 μM, respectively).

The docking studies are also not competitive. There is some major concern about the study, please justify them.

    Response: The Authors thank the reviewer to improve the manuscript. The entire docking studies have been redone by using another target protein, the newly synthesized compounds gave a competitive result with the new one in which compound 6h binding energy equal -10.8 kcal/mol with significant result comparing with the standard drug (-7.1 kcal/mol)

  1. Why did the authors have chosen the 1M17 PDB entry? 

Response: We thank the reviewer to raise this question. The entire docking studies have been redone by using 3W33 PDB

  1. There are many PDB entries available in the PDB in recent years. Have the authors reviewed them?

Response: We thank the reviewer to raise this question. The authors reviewed the most recent target protein available in the PDB like 3W2S, 3W32 and 3W33 however 3W33 has been selected due to its high X-ray resolution (1.70 Å)

  1. The PDB entry 1m17 contains one native ligand (AQ4). Did the authors perform a re-docking of it?

Response: We thank the reviewer for raising this question. The entire docking studies have been redone by using 3W33 PDB, which contains a native ligand (W19). In spite of the re-docking between 3W33 and W19, which was given a binding free energy equal to-10.8 kcal/mol, which is the same result as our compound 6h, we did not mention that in the manuscript to maintain consistency between the in vitro cytotoxicity evaluation and the molecular docking study, so as to have the same standard reference drug in both.

  1. Please provide the docking software details with the license and other details.

Response: We thank the reviewer asking this query. It was mentioned in section 3.2 with referring to the reference (45). Molecular Operating Environment (MOE) (2019) Montreal, Quebec, Canada: Chemical Computing Group Inc. https://www.chemcomp.com/ FEATURE moe chemcompd 2019.12 31-dec-2025 uncounted F782780B5A27 \HOSTID=ANY ISSUER=Lz0 TS_OK

  1. Molecular docking is not enough nowadays. It requires Molecular dynamics simulations. Why did the authors have not done MD simulation studies?

Response: We thank the reviewer for raising this question. MD is very useful for understanding the dynamic behavior of proteins or other biological macromolecules, from fast internal motions to slow conformational changes or even protein-folding processes. In our study, we concerned on a new series of aryl substituted novel pyrinine-1,3,4-thiadiazoles, and thiazoles compounds were synthesized from a starting compound 1-(3-cyano-4,6-dimethyl-2-oxopyridin-1(2H)-yl)-3-phenylthiourea through straightforward but efficient condensation methods with high yield. All the structures were confirmed through elemental and spectral analysis where the plausible mechanistic approach for their formation were discussed. All the structures show effectiveness towards the inhibition of human colon carcinoma (HCT-116) as well as hepatocellular carcinoma (HepG2) cell line through in-vitro evaluation confirmed by molecular docking study to determine how our candidate compounds may bind in the active site of a selected target and tries to predict how tightly it bind. This study attempts to mimic the process of bringing together a protein and a ligand to form a noncovalent complex, and to reveal the electrostatic and steric complementarity between the protein and ligand. Moreover, MDS methodologies have limitations including the high computational costs and approximations of molecular forces required, with constant improvements in both computer power and algorithm design

  1. The docking image quality is very poor and not understandable.

Response: We thank the reviewer asking this query. The whole docking studies changed, new images and figures have been inserted in high quality.

Reviewer 3 Report

Overall, the article is well written, but the authors have to make some critical improvements.

I. The introduction is poorly addressed. The authors have to update this section with newer references in accordance with their claims from lines 43-45. (it is not compulsory to remove old references – those references are good but obsolete) . I have some suggestions here – read and make adjustments, with a more extended discussion on them:

·         Concept of Hybrid Drugs and Recent Advancements in Anticancer Hybrids / https://doi.org/10.3390/ph15091071

·         Thiazoles: A Retrospective Study on Synthesis, Structure-activity Relationship and Therapeutic Significance / https://www.ijper.org/article/1737

·         An efficient One-pot three-component synthesis, Molecular docking, ADME and DFT predictions of new series Thiazolidin-4-one derivatives bearing a Sulfonamide moiety as potential Antimicrobial and Antioxidant agents / https://dx.doi.org/10.21608/ejchem.2022.104381.4819

·         5-arylidene(Chromenyl-methylene)-thiazolidinediones: Potential new agents against mutant oncoproteins K-Ras, N-Ras and B-Raf in colorectal cancer and melanoma / https://doi.org/10.3390/medicina55040085

·         Discovery of New 3,3-Diethylazetidine-2,4-dione Based Thiazoles as Nanomolar Human Neutrophil Elastase Inhibitors with Broad-Spectrum Antiproliferative Activity / https://doi.org/10.3390/ijms23147566

·         Synthesis and Antimicrobial Activity Screening of Piperazines Bearing N,N′-Bis(1,3,4-thiadiazole) Moiety as Probable Enoyl-ACP Reductase Inhibitors / https://doi.org/10.3390/molecules27123698

·         Antibacterial Evaluation and Virtual Screening of New Thiazolyl-Triazole Schiff Bases as Potential DNA-Gyrase Inhibitors / https://doi.org/10.3390/ijms19010222

·         Design, Synthesis, Antibacterial Evaluations and In Silico Studies of Novel Thiosemicarbazides and 1,3,4-Thiadiazoles / https://doi.org/10.3390/molecules27103161

·         Antioxidant activity and antibacterial evaluation of new thiazolin-4-one derivatives as potential tryptophanyl-tRNA synthetase inhibitors / https://doi.org/10.1080/14756366.2019.1596086

·         Synthesis, Molecular Docking Analysis and Biological Evaluations of Saccharide-Modified Thiadiazole Sulfonamide Derivatives / https://doi.org/10.3390/ijms22115482

II. Molecular docking protocol: target structure is a wrong choice in terms of X-ray structure quality – 1M17 has a very bad resolution (resolution of 2.60 Å is inappropriate for docking studies – please read “Chemoinformatics Approaches to Virtual Screening”, Editors: Alexandre Varnek & Alex Tropsha; also, see this discussion: https://www.researchgate.net/post/For-docking-what-are-the-criteria-to-choose-the-best-protein-structure-from-PDB-to-be-used-as-the-target/1). Therefore, the entire docking work needs to be redone, using one of the following structures:

ID            Method               Resolution          Chain         Positions

3W2S     X-ray                     1.90                     A             696-1022                             

3W32     X-ray                     1.80                     A             696-1022             

3W33     X-ray                     1.70                     A             696-1022

The best choice is 3W33 because it has the highest resolution.

Moreover, when you are using PDB site and structures from PDB, please fully comply with PDB citing policies: https://www.rcsb.org/pages/policies

III. Miscellaneous:

·         Figure 7 is distorted

·         Figure 8 is a very low quality image

·         Line 189: “Elementar vario LIII CHNS analyzer (German Made)”, please specify exactly the equipment and the producer, land (if Germany)

·         Why are some references colored in red (check lines: 163, 206)?

·         Line 330: “The details are provided in the supporting information file [44].” The supporting information file of the cited article or of this paper?

·         Line 352: a reference is needed for the method: “In a nutshell, the docking site was chosen using the dummy atoms method”.

·         Line 345 & Reference 42: incorrect citation for structure of 1M17! Correct citation from the web page of 1M17 (https://www.rcsb.org/structure/1M17) is:

Structure of the epidermal growth factor receptor kinase domain alone and in complex with a 4-anilinoquinazoline inhibitor.

Stamos, J., Sliwkowski, M.X., Eigenbrot, C.

(2002) J Biol Chem 277: 46265-46272

DOI: 10.1074/jbc.M207135200 (http://dx.doi.org/10.1074/jbc.M207135200)

Primary Citation of Related Structures: 

1M14, 1M17

Author Response

Dear reviewer! Many thanks for your time spending and efforts to review the manuscript.

  1. The introduction is poorly addressed. The authors have to update this section with newer references in accordance with their claims from lines 43-45. (it is not compulsory to remove old references – those references are good but obsolete) . I have some suggestions here – read and make adjustments, with a more extended discussion on them:

  • Concept of Hybrid Drugs and Recent Advancements in Anticancer Hybrids / https://doi.org/10.3390/ph15091071

Singh, A.K.; Kumar, A.; Singh, H.; Sonawane, P.; Paliwal, H.; Thareja, S.; Pathak, P.; Grishina, M.; Jaremko, M.; Emwas, A.-H.; Yadav, J.P.; Verma, A.; Khalilullah, H.; Kumar, P. Concept of hybrid drugs and recent advancements in anticancer hybrids. Pharmaceuticals 2022, 15, 1071.  

  • Thiazoles: A Retrospective Study on Synthesis, Structure-activity Relationship and Therapeutic Significance / https://www.ijper.org/article/1737

Sharma, S.; Devgun, M.; Narang, R.; Lal, S.; Rana, A.C. Thiazoles: a retrospective study on synthesis, structure-activity relationship and therapeutic significance. Indian Journal of Pharmaceutical Education and Research, 2022; 56, 646-666.

  • An efficient One-pot three-component synthesis, Molecular docking, ADME and DFT predictions of new series Thiazolidin-4-one derivatives bearing a Sulfonamide moiety as potential Antimicrobial and Antioxidant agents / https://dx.doi.org/10.21608/ejchem.2022.104381.4819

Hassan, S.; Abdullah, M.; Aziz, D. An efficient one-pot three-component synthesis, molecular docking, ADME and DFT predictions of new series thiazolidin-4-one derivatives bearing a sulfonamide moiety as potential antimicrobial and antioxidant agents. Egyptian J. Chem., 2022, 65, 133-146.

  • 5-arylidene(Chromenyl-methylene)-thiazolidinediones: Potential new agents against mutant oncoproteins K-Ras, N-Ras and B-Raf in colorectal cancer and melanoma / https://doi.org/10.3390/medicina55040085

Nastasă, C.; Tamaian, R.; Oniga, O.; Tiperciuc, B. 5-Arylidene(chromenyl-methylene)-thiazolidinediones: potential new agents against mutant oncoproteins K-Ras, N-Ras and B-Raf in colorectal cancer and melanoma. Medicina 2019, 55, 85.

  • Discovery of new 3,3-diethylazetidine-2,4-dione based thiazoles as nanomolar human neutrophil elastase inhibitors with broad-spectrum antiproliferative activity / https://doi.org/10.3390/ijms23147566

Donarska, B.; Świtalska, M.; Wietrzyk, J.; Płaziński, W.; Mizerska-Kowalska, M.; Zdzisińska, B.; Łączkowski, K.Z. Discovery of new 3,3-diethylazetidine-2,4-dione based thiazoles as nanomolar human neutrophil elastase inhibitors with broad-spectrum antiproliferative activity. Int. J. Mol. Sci. 2022, 23, 7566. 

  • Antibacterial Evaluation and Virtual Screening of New Thiazolyl-Triazole Schiff Bases as Potential DNA-Gyrase Inhibitors / https://doi.org/10.3390/ijms19010222

Nastasă, C.; Vodnar, D.C.; Ionuţ, I.; Stana, A.; Benedec, D.; Tamaian, R.; Oniga, O.; Tiperciuc, B. antibacterial evaluation and virtual screening of new thiazolyl-triazole schiff bases as potential dna-gyrase inhibitors. Int. J. Mol. Sci. 2018, 19, 222. 

  • Antioxidant activity and antibacterial evaluation of new thiazolin-4-one derivatives as potential tryptophanyl-tRNA synthetase inhibitors / https://doi.org/10.1080/14756366.2019.1596086

Stana, A.; Vodnar, D. C.; Marc, G.; Benedec, D.; Tiperciuc, B.; Tamaian, R.; Oniga, O. Antioxidant activity and antibacterial evaluation of new thiazolin-4-one derivatives as potential tryptophanyl-tRNA synthetase inhibitors.  J. Enzyme Inhibition and Med.Chem. 2019, 34, 898-908.

  • Synthesis and antimicrobial activity screening of piperazines bearing n,n′-bis(1,3,4-thiadiazole) moiety as probable enoyl-ACP reductase inhibitors / https://doi.org/10.3390/molecules27123698

Omar, A.Z.; Alshaye, N.A.; Mosa, T.M.; El-Sadany, S.K.; Hamed, E.A.; El-Atawy, M.A. Synthesis and antimicrobial activity screening of piperazines bearing n,n′-bis(1,3,4-thiadiazole) moiety as probable enoyl-ACP reductase inhibitors. Molecules 2022, 27, 3698.  

  • Design, Synthesis, Antibacterial Evaluations and In Silico Studies of Novel Thiosemicarbazides and 1,3,4-Thiadiazoles / https://doi.org/10.3390/molecules27103161

Janowska, S.; Khylyuk, D.; Andrzejczuk, S.; Wujec, M. Design, synthesis, antibacterial evaluations and in silico studies of novel thiosemicarbazides and 1,3,4-thiadiazoles. Molecules 2022, 27, 3161. 

  • Synthesis, Molecular Docking Analysis and Biological Evaluations of Saccharide-Modified Thiadiazole Sulfonamide Derivatives / https://doi.org/10.3390/ijms22115482

Zhang, Z.-P.; Zhong, Y.; Han, Z.-B.; Zhou, L.; Su, H.-S.; Wang, J.; Liu, Y.; Cheng, M.-S. Synthesis, molecular docking analysis and biological evaluations of saccharide-modified thiadiazole sulfonamide derivatives. Int. J. Mol. Sci. 2021, 22, 5482.

Response: All suggested references were added (see revised manuscript)

  1. Molecular docking protocol: target structure is a wrong choice in terms of X-ray structure quality – 1M17 has a very bad resolution (resolution of 2.60 Å is inappropriate for docking studies – please read “Chemoinformatics Approaches to Virtual Screening”, Editors: Alexandre Varnek & Alex Tropsha; also, see this discussion: https://www.researchgate.net/post/For-docking-what-are-the-criteria-to-choose-the-best-protein-structure-from-PDB-to-be-used-as-the-target/1). Therefore, the entire docking work needs to be redone, using one of the following structures:

ID            Method               Resolution          Chain         Positions

3W2S     X-ray                     1.90 Å                    A             696-1022                             

3W32     X-ray                     1.80 Å                    A             696-1022             

3W33     X-ray                     1.70 Å                    A             696-1022

The best choice is 3W33 because it has the highest resolution.

Moreover, when you are using PDB site and structures from PDB, please fully comply with PDB citing policies: https://www.rcsb.org/pages/policies

Response: The Authors thank and appreciate the reviewer for his kind notes, information's and advises to improve our manuscript. The entire docking studies have been redone by using 3W33 PDB which is the best choice as a protein target. Authors obey PDB citing Policies.

III. Miscellaneous:

  • Figure 7 is distorted

Response: We thank the reviewer asking this query. The whole docking studies changed, new images and figures have been inserted in high quality.

  • Figure 8 is a very low quality image

Response: We thank the reviewer asking this query. The whole docking studies changed, new images and figures have been inserted in high quality.

  • Line 189: “Elementar vario LIII CHNS analyzer (German Made)”, please specify exactly the equipment and the producer, land (if Germany)

Response: (Elementar Analysensysteme GmbH, Langenselbold, Germany)

  • Why are some references colored in red(check lines: 163, 206)?

Response: the colored references in the previous version belong to molecular docking and have been colored in black in the revised version

  • Line 330: “The details are provided in the supporting information file [44].” The supporting information file of the cited article or of this paper?

Response: this paper

  • Line 352: a reference is needed for the method: “In a nutshell, the docking site was chosen using the dummy atoms method”.

Response: We thank the reviewer asking this query. The reference was mentioned in the manuscript, reference no 53

  1. N. Grima, R. Gatt, T. G. C. Bray, A. Alderson & K. E. Evans (2005) Empirical modelling using dummy atoms (EMUDA): an alternative approach for studying “auxetic” structures, Molecular Simulation, 31:13, 915-924, DOI: 10.1080/08927020500401121

  • Line 345 & Reference 42: incorrect citation for structure of 1M17! Correct citation from the web page of 1M17 (https://www.rcsb.org/structure/1M17) is:

Structure of the epidermal growth factor receptor kinase domain alone and in complex with a 4-anilinoquinazoline inhibitor.

Stamos, J., Sliwkowski, M.X., Eigenbrot, C. (2002) J Biol Chem 277: 46265-46272

DOI: 10.1074/jbc.M207135200 (http://dx.doi.org/10.1074/jbc.M207135200)

Primary Citation of Related Structures: 

Response: We thank the reviewer asking this query. The entire docking studies have been redone by using 3W33 PDB (https://www.rcsb.org/structure/3W33) and its citation changed by the correct one which is: Design and synthesis of novel pyrimido[4,5-b]azepine derivatives as HER2/EGFR dual inhibitors. Kawakita, Y., Seto, M., Ohashi, T., Tamura, T., Yusa, T., Miki, H., Iwata, H., Kamiguchi, H., Tanaka, T., Sogabe, S., Ohta, Y., Ishikawa, T. (2013) Bioorg Med Chem 21: 2250-2261

DOI: 10.1016/j.bmc.2013.02.014

Primary Citation of Related Structures:  3W32, 3W33

Round 2

Reviewer 1 Report

authors considered all comments so the article is accepted 

Author Response

Dear reviewer!

Many thanks for your time spending and efforts to review the manuscript.  

Reviewer 2 Report

Although the revised version of the manuscript improved a lot, however still few questions are not satisfactorily answered. Additionally, there are some major points that need further clarification.

Question not answered appropriately: Interestingly, the authors have not understood the significance of question 14. Or reluctantly, ignored it. It should be clarified.

1. The most recent PDBs are 5U8L, 5UG8, etc. with low resolution. Please take a look at the below link and come up with a robust answer for PDB selection.

https://www.uniprot.org/uniprotkb/P00533/entry#structure

2. Abstract needs to be more clear.

3. I didn't find any justification about molecular docking study importance in the introduction section.

4. Please provide the images and data for the anticancer activities.

5. Image quality is still pathetic and unable to read/understand.

6. Provide the NMR data of the synthesized compounds.

7. Protein and ligand preparation is still missing.

8. Which algorithm was used for docking?

9. 100 ns molecular dynamics (MD) simulation is very common now a days and any researcher can afford this so the excuse is not acceptable for MD simulation study. 

10. Conculsion is still chaotic.

11. Redocking proves the docking study is perfect and it can be carried for test compounds also hence it must be in the manuscript.

Author Response

Dear reviewer!

Many thanks for your time spending and efforts to review the manuscript. Your comments and suggestion were incorporated into the text.

  1. The most recent PDBs are 5U8L, 5UG8, etc. with low resolution. Please take a look at the below link and come up with a robust answer for PDB selection.

https://www.uniprot.org/uniprotkb/P00533/entry#structure

 Response: The authors thank reviewer for providing comment and the link of the most recent PDBs like 5U8L and 5UG8 however the selection of 3W33 was based on:

  • Suggestion of the reviewer 3 who offers many PDBs with advising 3W33
  • 3W33 is still modern (2013) with high X-ray resolution (1.7 A0) as well as high quality assessment
  • 3W33 contains one native ligand (W19) which is like our ligands based on similarity database (https://pubchemdocs.ncbi.nlm.nih.gov/structure-search)
  • This similarity offer specifying the residues of a pocket and selecting the site points leading to great binding affinities

  1. Abstract needs to be more clear.

Response: Abstract was revised and updated to make it clearer

  1. I didn't find any justification about molecular docking study importance in the introduction section.

Response: The following paragraph justifies the molecular docking study:

“The present report aims to elaborate about the new series of thiadiazole-pyridines as well as thiazole-pyridines that might have cytotoxic effects via inhibition of protein Epidermal Growth Factor Tyrosine Kinase receptor (EGFR TK) which plays an essential mediating role in cell proliferation, angiogenesis, apoptosis, and metastatic spread compared with reference drugs”

  1. Please provide the images and data for the anticancer activities.

Response: the images and data of the anticancer activities for the most active compound 4h was attached in supporting file

  1. Image quality is still pathetic and unable to read/understand.

Response: High quality image was inserted with publication resolution 600 dpi however, Table 2 indicates the interacted residues, types of interaction and the distance of interactions between our candidates and selected protein.

  1. Provide the NMR data of the synthesized compounds.

Response: the NMR data of the synthesized compounds were attached in the supporting file

  1. Protein and ligand preparation is still missing.

Response: For Ligand: in section 3.2, it was mentioned that "The newly prepared derivatives were placed into the MOE window, where they were treated to partial charge addition and energy minimization [52, 53]. The produced compounds also were placed into a single database with the Harmine and saved as an MDB file, which was then uploaded to the ligand icon during the docking process".

For Protein: in section 3.2, it was mentioned that " The Protein Data Bank was used to generate an X-ray of the targeted Epidermal Growth Factor Receptor Tyrosine Kinase Domain (EGFR TK) (https://www.rcsb.org/structure/3W33) [48]. Furthermore, it was readied for the docking process by following the previously detailed stages [54, 55]. Additionally, the downloaded protein was error-corrected, 3D hydrogen-loaded and energy-minimized [56, 57]".

  1. Which algorithm was used for docking?

Response: in section 3.2, it was mentioned that "Triangle matcher and London dG, were chosen. The stiff receptor was used as the scoring method, and the GBVI/WSA dG was used as the refining method, respectively, to select the top 10 poses out of a total of 100 poses for each docked molecule [60, 61]".

  1. 100 ns molecular dynamics (MD) simulation is very common now a days and any researcher can afford this so the excuse is not acceptable for MD simulation study. 

Response: Our aim in this study is the synthesis of new thidiazoles and thiazoles candidates with in vitro cytotoxic evaluation against human colon carcinoma and hepatocellular carcinoma cell lines. To further understand and rationalize the inhibitory effect of our candidates against EGFR TK, docking studies were performed, which suggested that most of the synthesized compounds have similar binding mode with target crystal structure. Therefore, this study presents compounds 4h as potential promising candidate as EGFR TK inhibitor.

On the other hand, MD is very useful for understanding the dynamic behavior of proteins or other biological macromolecules, from fast internal motions to slow conformational changes or even protein-folding processes which isn't our aim in this study.

  1. Conclusion is still chaotic.

Response: Conclusion was revised and updated

  1. Redocking proves the docking study is perfect and it can be carried for test compounds also hence it must be in the manuscript.

Response: Done with thanks. Figures 10 and 11

Reviewer 3 Report

The manuscript has been greatly improved!

However, the docking protocol is missing the control element: re-docking of the co-crystalized ligand from X-ray structure of the target (3W33) - namely W19: 
https://www.rcsb.org/ligand/W19 

Is compulsory that the authors proceed with re-docking of W19.

Author Response

Dear reviewer!

Many thanks for your time spending and efforts to review the manuscript. Your comments and suggestion were incorporated into the text.

However, the docking protocol is missing the control element: re-docking of the co-crystalized ligand from X-ray structure of the target (3W33) - namely W19: 
https://www.rcsb.org/ligand/W19 

Is compulsory that the authors proceed with re-docking of W19.

Response: Done with great appreciation. Figures 10 and 11